# Quercetin’s Dual Mode of Action to Counteract the Sp1-miR-27a Axis in Colorectal Cancer Cells

**DOI:** 10.3390/antiox12081547

**Published:** 2023-08-02

**Authors:** Emanuele Fosso, Manuela Leo, Livio Muccillo, Vittorio Maria Mandrone, Maria Chiara Di Meo, Annamaria Molinario, Ettore Varricchio, Lina Sabatino

**Affiliations:** Department of Sciences and Technologies, University of Sannio, Via Francesco de Sanctis, 82100 Benevento, Italy; emafosso@unisannio.it (E.F.); manleo@unisannio.it (M.L.); muccillo@unisannio.it (L.M.); vittorio.mandrone13@gmail.com (V.M.M.); mardimeo@unisannio.it (M.C.D.M.); molinario@unisannio.it (A.M.); etvarric@unisannio.it (E.V.)

**Keywords:** quercetin, colorectal cancer cells, miRNAs, miR-27a, miR-23a, miR-24-2, Sp1, ZBTB10, proteasome degradation

## Abstract

Quercetin (Qc) inhibits cell proliferation and induces apoptosis in a variety of cancer cells. The molecular mechanism of action has not been fully elucidated; however, interplay with some miRNAs has been reported, specifically with miR-27a, an onco-miRNA overexpressed in several malignancies. Here, we show that Qc reduces cell viability and induces apoptosis in HCT116 and HT-29 colon cancer cells, by upregulating negative modulators of proliferation pathways such as Sprouty2, PTEN and SFRP1. These are targets of miR-27a whose high expression is reduced by Qc. Moreover, miR-23a, and miR-24-2, the two other components of the unique gene cluster, and the pri-miRNA transcript are reduced, evoking a transcriptional regulation of the entire cluster by Sp1. Mechanistically, we show that Qc is rapidly internalized and localizes in the nucleus, where it likely interacts with Sp1, inducing its proteasomal degradation. Sp1 is further repressed by ZBTB10, an Sp1 competitor for DNA binding that is an miR-27a target and whose levels increase following Qc. *SP1* mRNA is also reduced, supporting the regulation of its own gene transcription. Finally, Sp1 knockdown elicits the impaired transcription of the entire cluster and the upregulation of the miR-27a targets, phenocopying the effects of Qc. Through this dual mode of action, Qc counteracts the protumoral Sp1-miR-27a axis, opening the way for novel therapies based on its association as neoadjuvant with known anticancer treatments.

## 1. Introduction

Quercetin [2-(3, 4-dihydroxy-phenyl)-3, 5, 7-trihydroxy-4H-1-benzopyran-4-one] (Qc) belongs to the flavonol subclass of flavonoids, a family of compounds ubiquitously found in plants, plant food sources, and in the human diet [1,2,3].

Flavonoids are part of a larger family of natural compounds, polyphenols, which modulate various signaling pathways by influencing the expression of genes involved in cell-cycle regulation, differentiation, and apoptosis [4,5]. Thus, Qc displays a high antioxidant, anti-inflammatory, and antiproliferative activity. Quercetin’s anti-inflammatory activities have been explored in many in vitro and in vivo studies [6] and tested on cancer cells which exhibit a more pronounced response than normal and nontransformed cells [7]. In line with these observations, Qc proved to inhibit cell proliferation and angiogenesis while it induced apoptosis in a variety of cancers’ cells in vitro, including colorectal cancer (CRC) [8,9,10]. These activities were confirmed in diverse tumor models in vivo, along with an additional anti-metastatic potential, supporting its role as an anticancer agent.

Consistent with these data, Qc has been added to the diet and positively associated with improved outcomes in cancer treatments [10]. Despite this wealth of data, the exact mechanisms through which Qc exerts its effects have not been completely elucidated. Interplay with some microRNAs has also been reported [11,12,13,14].

MicroRNAs (miRNAs) are a family of endogenous noncoding RNA molecules of about 18–23 nucleotides in length, that regulate gene expression at the post-transcriptional level. They recognize cognate seed sequences in the coding or the 3′ untranslated region of target mRNAs, leading to degradation or translational repression [15,16]. Over 2000 microRNAs have been discovered so far and about 60% of all genes coding for human proteins can be targeted. Consistently, miRNAs play important roles in a myriad of processes, including neuronal metabolism, cell death, cell proliferation, hematopoietic differentiation, and immunity [15,16]. miRNAs also influence the tumorigenic process modulating the expression of oncogenes or tumor suppressor genes, or acting as onco-miRNAs or tumor suppressor miRNAs by themselves [17,18].

Qc has been reported to inhibit cell growth in colon and renal cancer cells, in association with resveratrol and hyperoside through the inhibition of miR-27a [19,20]. This miRNA is overexpressed in a series of tumors and tumor cell lines and acts as an onco-miRNA [21,22,23]. We and others showed that in CRC cells, miR-27a stimulates proliferation and invasiveness by activating the RAS, AKT, and the Wnt/β-catenin signaling pathways [22,24]. miR-27a has also been shown to upregulate the expression of specificity factor 1 (Sp1), a ubiquitous transcription factor (TF), through targeting and downmodulating ZBTB10, a protein that acts as an Sp1 repressor [25,26,27].

Sp1 is a member of a family of TFs with critical roles in embryonic development, as documented by the early lethality or multiple developmental defects in *Sp* genes knockout mice [28,29]. Sp1 is expressed at low levels in multiple adult tissues and organs and further decreases in expression or DNA binding with increasing age [25,26]. In contrast, Sp factors are overexpressed in tumors stimulating growth-related and cell-cycle control molecules, oncogenes, and angiogenesis-related factors, while they inhibit apoptosis and tumor suppressor genes. For these reasons, Sp TFs are considered negative prognostic factors for cancer patients’ survival [25,26,30].

Based on these data, we sought to investigate the molecular mechanisms underlying the Qc effects in an in vitro CRC cell model system. We confirm that Qc induces apoptotic cell death and reduces cell viability by unleashing some of the most relevant inhibitors of the major cell proliferation pathways. These inhibitors are targets of miR-27a that, in turn, is reduced by Qc along with miR-23a and miR-24-2, the other two members of the cluster which miR-27a belongs to. The transcription of the entire cluster is impaired by Qc through the Sp1 repression by ZBTB10, another miR-27a target. Qc not only restrains this self-sustaining regulatory loop but also drives the Sp1 proteasomal degradation, further impairing the Sp1 and miR-27a transcription. Finally, Sp1 knockdown recapitulates the effects of Qc, underscoring the central role played by this TF.

## 2. Material and Methods

### 2.1. Preparation of Quercetin Stock Solution and Working Solution

A commercially available Qc (Merck-Sigma-Aldrich, St. Louis, MI, USA, Catalogue number 337951 with >95% purity by HPLC) was dissolved in dimethyl sulfoxide (DMSO) (ChemCruz, Santacruz Biotechnology, Dallas, TX, USA) to prepare a stock solution of 30 mg/mL corresponding to 99.26 mM, within the limits of solubility in the DMSO solvent.

### 2.2. Cell Culture and Treatments

HT116 and HT-29 human CRC-derived cell lines were purchased from American Type Culture Collection (ATCC, Manassas, VA, USA) and cultured in RPMI or DMEM medium (Life Technologies, Carlsbad, CA, USA), respectively, supplemented with 10% *v*/*v* fetal bovine serum (Life Technologies), 2 mM L-glutamine, 100 U mL^−1^ penicillin/streptomycin at 37 °C under 5% CO_2_. The experiments were carried out using the doses 50 µM or 150 µM for 24 h in HCT116 and HT-29 cells, respectively. Control cells were treated with equivalent amounts of DMSO, the vehicle in which Qc was dissolved. MG132 (Calbiochem-Merck, Darmstadt, Germany) was administered alone or in combination with Qc at the concentration of 10 µM for the last 5 h of the treatment.

### 2.3. Cell Viability Assay and Calculation of IC_50_ Value

HCT116 and HT-29 cells from three different freezing stocks were seeded in 96-well plate in complete medium and left overnight in the incubator to enhance adherence. The next day, the medium was replaced, and the cells exposed to increasing doses of Qc, from 0 to 900 µM. Viable cells were measured with Prestoblue™ Cell Viability Reagent (Thermo Fisher Scientific, Waltham, MA, USA) and the fluorescence intensity (FI) detected with a Tecan Infinite-Pro 200 plate reader using an excitation at 540 nm and an emission at 595 nm. GraphPad Prism software (GraphPad software, Inc., version 8.1.1, San Diego, CA, USA) was used for the calculation of the IC_50_ value. Data were transformed to a log scale and normalized, setting the dose 0 µM as 100% and 900 µM as 0%.

### 2.4. Flowcytometry Analysis

In total, 6 × 10^5^ HCT116 or 7 × 10^5^ HT-29 cells were seeded in 60 mm plates and incubated overnight at 37 °C under 5% CO_2_. The next day, cells were treated with Qc at the concentrations indicated above for 24 h. At the end of the treatment, cells were harvested, including dying cells floating in the medium, washed with PBS, and processed with the FITC Annexin V Apoptosis Detection Kit I (556547, Becton Dickinson—BD, San Jose, CA, USA). Cell death was evaluated by BD FACS Celesta and data acquired with FACS Diva software, version 9.2 (BD).

### 2.5. Western Blotting Analysis

A Western blot analysis was carried out as previously reported [31]. Briefly, at the end of each treatment, cells were harvested and lysed in an ice-cold lysis buffer containing 25 mM Tris-HCl, pH 7.5, 150 mM NaCl, 2 mM EDTA, 1% Triton X-100, 1% sodium deoxycholate, 0.1% SDS, and a cocktail of protease and phosphatase inhibitors (Roche, Basel, Switzerland). After heating at 95 °C for 5 min, 30 µg of protein extracts for each sample were loaded on nonreducing polyacrylamide gels, transferred onto PVDF membranes, and blocked with 5% nonfat dry milk for 1 h at room temperature. Subsequently, the membranes were incubated with primary antibodies overnight at 4° C, such as PARP (#9542), phospho-ERK 1/2 T202/Y204 (#4370), ERK 1/2 (#9102), phospho-AKT T308 (#2965), AKT (#9272), phospho-mTOR S2448 (#2971), and mTOR (#2983) purchased from Cell Signaling Technology (Beverly, MA, USA); β-catenin (610153) from BD; Sp1 (GTX636836) from Genetex (Irvine, CA, USA); ZBTB10 (ab117786) from Abcam (Cambridge, UK); β-Actin (F-3022) and α-Tubulin (T-5168) from Merck-Sigma-Aldrich. Secondary antibodies, either antimouse or antirabbit antibodies conjugated with horseradish peroxidase, were incubated 1 h at room temperature. Bands were detected by Clarity western ECL Substrate (#1705061, BIO-RAD, Hercules, CA, USA) using the ChemiDoc apparatus (BIO-RAD). Bands were quantitated by using ImageLab software, version 6.1.0 (BIO-RAD). Blots were cut and probed with antibodies directed against different proteins, including β-Actin and α-Tubulin, used as loading control. To detect proteins with similar molecular weight, some PVDF membranes were subjected to a mild stripping, according to the Abcam protocol.

### 2.6. RNA Extraction and Quantitative Real-Time PCR Analysis

Total RNA was extracted using TRIZOL^®^ Reagent (Invitrogen, Carlsbad, CA, USA) according to the manufacturer’s instructions. Reverse transcription was performed by SuperScript II Reverse Transcriptase (Thermo Fisher, Life Technologies) and cDNA used for quantitative real-time PCR (qRT-PCR) performed with PowerUp™ SYBR™ Green Master Mix (A25742 Applied Biosystems, Waltham, MA, USA). 18S RNA was used as reference for normalization.

cDNA from miRNAs was produced using miRNA RT (Qiagen, Hilden, Germany) or miRCURY LNA RT Kit (Qiagen); qRT-PCR was performed with SYBR Green (Qiagen) or miRCURY LNA Probe PCR Kit (Qiagen), respectively. U6 RNA expression was used as reference for data normalization. In each case, the QuantStudio 5 instrument (Applied Biosystems) was used to collect data and the fold change calculated by applying the formula below:

Fold change = 2^−ΔΔCT^, where −ΔΔCT = −(ΔCT_Goi_ − ΔCT_Ref_), Goi = gene of interest, Ref = reference gene

The sequences of the used primers and the relative annealing temperatures are reported here:

*SPRY2* (accession number: NM_005842).

*SPRY2* FW: CATGGGTGTCATGTCCCTCTT.

*SPRY2* RV: GCCTGTTAACCCGGTCATAACA.

T = 60 °C.

*PTEN* (accession number: NM_000314).

*PTEN* FW: CACACGACGGGAAGACAAGTTC.

*PTEN* RV: CCTCTGGTCCTGGTATGAAGAATG.

T = 62 °C.

*SFRP1* (accession number: NM_003012).

*SFRP1* FW: TTGAGGAGAGCACCCTAGGC.

*SFRP1* RV: TGTGTATCTGCTGGCAACAGG.

T = 60 °C.

*FBXW7* (accession number: NM_033632).

*FBXW7* FW: TGCAAAGTCTCAGAATATACA.

*FBXW7* RV: ATTTCTCTGGTCCACTCCAGC.

T = 58 °C.

*ANNEXIN1* (accession number: NM_000700).

*ANNEXIN1* FW: TAAGCGAAACAATGCACAGC.

*ANNEXIN1* RV: CAAAGCGTTCCGAAAATCTC.

T = 55 °C.

*18S* (accession number: NR_146151.1).

*18S* FW: GGGAGCCTGAGAAACGGC.

*18S* RV: GGGTCGGGAGTGGGTAATTT.

T = 60 °C.

*Pri-miR-23a~27a∼24-2 cluster* (accession number: NC_000019).

*Pri-miR-23a~27a∼24-2 cluster* FW: ATCACATTGCCAGGGATTTCCA.

*Pri-miR-23a~27a∼24-2 cluster* RV: GATTCTGAGTCCTCATCTCTGCT.

T = 62 °C.

*SP1* (accession number: NM_138473).

*SP1* FW: TCACCTGCGGGCACACTT.

*SP1* RV: CCGAACGTGTGAAGCGTT.

T = 60 °C.

*ZBTB10* (accession number: NM_023929).

*ZBTB10* FW: GGACCCGCAACTACAAGAAA.

*ZBTB10* RV: CCTCTGATGGTATTTCTGACTCCT.

T = 62 °C.

### 2.7. Conformational Analysis of miR-23a~27a∼24-2 Cluster Structure

The secondary structure of the miR-23a∼27a∼24-2 cluster was accomplished by using RNA structure v 6.4 software, as previously described [32]. The algorithm evaluates both the free energy at 37 °C and enthalpy changes to predict the RNA transcripts conformation stability.

### 2.8. Sp1 ChIP-seq Analysis from ENCODE Dataset

Sp1 ChIP-seq analysis was obtained by querying the HCT116 cells dataset available from the ENCODE consortium (https://www.encodeproject.org/experiments/ENCSR000BSF/) (accessed on 12 April 2023). The peak calls on the *SP1* promoter were determined following the ENCODE pipeline. Briefly, putative signals were mapped at the *SP1* binding sites and reported as fold/control at each position and as *p* value to reject the null hypothesis that a signal similar to that of the control was present at that location. Peak calls are significant when they exceed the threshold of 2% of IDR (irreproducible discovery rate), a reproducibility measure identified by comparing the results of replicate experiments.

### 2.9. Immunofluorescence and Microscopy

To assess Qc internalization, HCT116 cells were seeded on glass coverslips and incubated overnight at 37° C under 5% CO_2_ to enhance attachment. Cells were treated with 50 µM of the compound from 5 min to 24 h, fixed in paraformaldehyde (PFA) 3%, permeabilized with 0.1% Triton-X-100, and exposed to DAPI for nuclear staining. The coverslips were subjected to several washes in PBS at each step. Qc autofluorescence was detected in the FITC channel. For the Sp1 localization, HCT116 cells, grown on coverslips overnight, were treated with 50 µM Qc for 24 h, fixed and permeabilized as above, blocked in BSA solution, and incubated with Sp1 (GTX636836) and α-Tubulin (T-5168) antibodies. Secondary antibodies conjugated to Alexa Fluor 546 or 488 were used, respectively. DAPI was used to stain nuclei. The coverslips were subjected to several washes in PBS at each step. To quantify Sp1 fluorescence, a significant number of cells (>100 in at least 10 different and distant microscopic fields) was captured with the same setting and then processed in QuPath v.0.4.3 [33]. Identification was performed using QuPath’s built-in “Cell detection” algorithm as reported, and for all acquired cells, TRITC signals were registered, exported, and analyzed in Graphpad software. Data are shown as the mean fluorescence intensity (MFI).

Images were acquired by using the fluorescence microscope Nikon Eclypse Ti (Amstelveen, The Netherlands). NIS Elements AR v.5.20.00 software was used for the image acquisition.

### 2.10. Transfection of SP1 siRNAs

HCT116 and HT-29 cells were seeded in 6-well plates and left overnight at 37 °C under 5% CO_2_ to enhance adhesion. Three different siRNAs directed against *SP1* mRNA were purchased from IDT (Coralville, IA, USA) and transfected by using RNAi MAX (Thermo Fisher, Life Technologies), according to the manufacturer’s instructions. After 48 h, cells were harvested and processed for RNA and protein extraction as reported above.

### 2.11. Statistical Analysis

All data are shown as mean ± SD of at least three independent experiments. Statistical significance was calculated using an ANOVA with Dunnett’s post-test whose significance is shown as # *p* < 0.05, ## *p* < 0.01, ### *p* < 0.001, or #### *p* < 0.0001, or a *t*-test with significance shown as * *p* < 0.05, ** *p* < 0.01, *** *p* < 0.001, or **** *p* < 0.0001. The statistical analysis was conducted using GraphPad Prism software, Inc., version 8.1.1, San Diego, CA, USA.

## 3. Results

### 3.1. Quercetin Affects Cell Viability and Promotes Cell Death in HCT116 Cells

In order to investigate the mechanisms underlying the effects of Qc, we first established the best dose that influenced the viability of HCT116 cells, with no toxicity. This cell line is derived from a CRC and has previously been used as a reliable cell model to test the effects of natural compounds [34,35,36,37]. Thus, we treated the cells with increasing doses of Qc from 0 to 900 µM for 24 h and evaluated their viability by the Presto Blue™ assay with respect to cells treated with DMSO, the vehicle in which Qc was dissolved. We observed a dose-dependent response (Figure 1A), and on this basis, we calculated the inhibitory concentration 50 (IC_50_), i.e., the dose that inhibits cell growth by 50%, which is a good compromise to observe biological effects. We found that the IC_50_ was 54.04 µM at 24 h (Figure 1B), so we used the concentration of 50 µM and 24 h treatment in all subsequent experiments as the best conditions.

To demonstrate that the reduction in cell viability was due to the activation of cell death programs, we carried out an apoptosis assay by flow cytometry. Qc, administered at 50 µM for 24 h, increased by about twofold the percentage of apoptotic and preapoptotic cells compared to vehicle-treated cells (Figure 1C). We confirmed the induction of apoptosis by a Western blot analysis of the cleaved form of PARP that was about twofold higher than vehicle-treated controls (Figure 1D). The cleavage of PARP is an event triggered by activated caspases [36].

Cell viability is also reduced by affecting cell proliferation. To test this possibility, in cells treated as above, we analyzed the three major pathways active in cancer cells: RAS/ERK, AKT/mTOR, and Wnt/β-catenin [37,38,39]. All three signaling pathways were downregulated, as the amounts of phosphorylated ERK and AKT, as well as that of β-catenin, were reduced by about 50% in treated vs. vehicle-treated HCT116 cells as assessed by a Western blot (Figure 2). The effect on the AKT signaling was confirmed by a 30% reduction in mTOR phosphorylation, a pathway that is strictly connected with AKT (Figure 2C).

### 3.2. Quercetin Regulates miR-27a Expression in HCT116 Cells

These data support that Qc significantly impairs the proliferation pathways active in tumor cells. This can be achieved by influencing either positive or negative modulators of the three above-mentioned signaling pathways; specifically, Sprouty2, PTEN, and SFRP1 negatively regulate the RAS/ERK, PI3K/AKT, and Wnt/β-catenin pathways [24,40,41], respectively, and we show here that the low basal levels of the corresponding mRNAs increased upon treatment with Qc (Figure 3A–C).

Interestingly, all three modulators are targets of miR-27a, an onco-miRNA in CRC tissues and cells [22,42]. We and others have shown that miR-27a-3p (henceforth indicated as miR-27a) is overexpressed in HCT116 cells and, consequently, a series of specific targets are downregulated. In addition to *SPRY2, PTEN,* and *SFRP1* mentioned above, two unrelated targets, *ANX1* and *FBXW7* mRNAs [22], were also upregulated (Figure 3D,E), supporting that the major effects attained by Qc occur through an miR-27a tuning down.

The gene coding for miR-27a belongs to a cluster located within an intergenic region of chromosome 19p13.22 that is transcribed as a single, polycistronic unit also containing miR-23a and miR-24-2, the two other members [43,44,45] (Figure 4A). To investigate the role of miR-27a, we analyzed its expression by qRT-PCR and found a reduction of about 50% following Qc exposure (Figure 4B). Despite the fact that the three miRNAs are contained in a single transcript, the levels of each of them can vary due to post-transcriptional processing [44,45]. Thus, we analyzed the expression of the other two miRNAs of the miR-23a~27a~24-2 cluster by qRT-PCR and found that they were both high in basal conditions and reduced (about 40%) following treatment with Qc with respect to vehicle-treated cells (Figure 4C,D). We further investigated the miRNA biosynthetic process by evaluating the amount of a 400-nucleotide-long RNA fragment comprised in the pri-miRNA. The primitive transcript of the cluster is more than 2000-nucleotide-long, harbors a CAP at its 5′ end and a putative polyadenylation signal at the 3′ end [43,44,45]. We showed that the pri-miRNA significantly diminished (about 40%) in HCT116 cells treated with the flavonol compared to vehicle-treated cells, as determined by qRT-PCR (Figure 4E). This indicates that Qc negatively regulates the transcription of the entire miR-23a~27a~24-2 cluster.

### 3.3. Quercetin Regulates the Expression of the miR-23a~27a~24-2 cluster in HCT116 Cells through Sp1

We inspected the 5′ flanking promoter region of the cluster and recognized it lacked a canonical TATA box and other promoter elements, while it was enriched in CpG and GC motifs, putative binding sites for the TF Sp1, up to −800 bp upstream of the transcription start site (TSS). Presumptive binding sites for other TFs such as YY1, C/EBP, AP2, and MYC were also identified, suggesting that they may influence the expression of the cluster [43,46]. We focused on Sp1 as it plays a crucial role in the transcription of genes lacking a canonical TATA box, such as the present one [25,26] (Figure 5A); in fact, Sp1 has been reported to bind to this promoter, influencing miR-27a expression in breast and laryngeal cancer cells [47,48]. Specifically, the two Sp1 binding sites between −600 and −400 bp upstream of the TSS are necessary elements for the transcription of the entire cluster, as their deletion reduces gene expression [43,47,48].

To support the correlation with miR-27a, we assessed Sp1 in HCT116 cells treated with Qc and found that the protein was 40% lower than vehicle-treated cells, as determined by a Western blot (Figure 5B, left panel). Interestingly, *SP1* mRNA also diminished, as documented by the qRT-PCR analysis, indicating that Qc affects the transcription of the corresponding gene (Figure 5B, right panel). The inspection of the human *SP1* promoter showed that it lacked a canonical TATA box, contained binding sites for AP2, C/EBP, MYC, NFY, and YY1, and was enriched in GC elements distributed over a long fragment of the 5′ flanking region [49]. This promoter architecture is reminiscent of that of the miR-23a~27a~24-2 cluster and suggests that Sp1 can bind and activate the transcription of its own gene in a positive regulatory loop (Figure 5C). This possibility is supported by the ChiP-seq data in HCT116 cells available in the ENCODE dataset (https://www.encodeproject.org/experiments/ENCSR000BSF/) (accessed on 12 April 2023) showing a strong binding for Sp1 in the region from -330 to -270 bp upstream of the TSS.

The interplay between Sp1 and miR-27a involves another factor, ZBTB10, a member of a zinc finger protein family that binds to the same GC motifs and competitively displaces Sp1 [27]. This is also a target of miR-27a, and consistently, ZBTB10 protein increased after Qc treatment by about twofold with respect to vehicle-treated cells (Figure 5D, left panel); *ZBTB10* mRNA levels also increased after treatment, suggesting that miR-27a induces a degradation of the corresponding mRNA, as for the other targets analyzed (Figure 5D, right panel). All together, these results indicate that Sp1 activates the transcription of miR-27a that, in turn, silences crucial negative factors of cell growth stimulating proliferation. ZBTB10 is an additional target, and its reduction modulates Sp1 function in an inverse regulatory loop. Qc disrupts this axis via reducing Sp1 and miR-27a expression, impairing cell vitality and proliferation, and activating cell death.

### 3.4. Quercetin Also Exerts Anticancer Effects in HT-29 Cell Line

To verify that the effects exerted by Qc were not restricted to HCT116 cells but were more general and reproduced in other CRC-derived cell lines, we performed similar experiments in the HT-29 cell line. These cells have previously been tested as responsive to Qc [19], have different genetic and epigenetic landscapes, and more importantly, express lower levels of miR-27a than HCT116, as we previously reported [22]. In this case as well, we exposed the cells to increasing doses of Qc (from 0 to 900 µM) for 24 h and measured the viability with the Presto Blue assay™. We detected a dose-dependent response especially between 100 and 300 µM (Appendix A), and on the basis of these values, we calculated the IC_50_ that turned out to be 157.6 µM (Appendix A). The 150 µM concentration was thus employed in all subsequent experiments. Next, we evaluated that the reduced viability was associated with an increase in apoptosis by measuring the percentage of pre- and apoptotic cells in flow cytometry using the annexin V and propidium iodide assay. The number of cells undergoing apoptosis was almost doubled by Qc with respect to vehicle-treated cells, as illustrated in Appendix A. This result was confirmed by assessing the amount of the cleaved form of PARP as an additional apoptotic marker in a Western blot analysis of protein extracts from Qc and vehicle-treated cells (Appendix A). We then analyzed the proliferation pathways influenced by Qc by checking the phosphorylation levels of ERK and AKT, the two executors of the growth factor/tyrosine kinase receptor/RAS or PI3K/AKT signaling, respectively. In both cases, p-ERK and p-AKT were reduced upon exposure to Qc by almost 40%, confirming its role in impairing cell proliferation in this cell context as well (Appendix A, left panels). As reported above, the growth signaling were downmodulated by Sprouty2 and PTEN, which in turn are under the control of miR-27a. Exposure to Qc reduced miR-27a and increased *SPRY2* and *PTEN* mRNAs (Appendix A, right panels). Consistently, miR-23a and miR-24-2 levels also diminished vs. vehicle-treated cells, indicating that Qc controls the expression of the entire miR-23a~27a~24-2 cluster (Appendix A).

Furthermore, the treatment significantly reduced Sp1 expression levels, both at the protein and mRNA levels (Appendix A) and, in parallel, promoted an increase (about 40%) in both ZBTB10 mRNA and protein (Appendix A). Collectively, in HT-29 cells, Qc influences the same pathways recapitulating the effects reported in HCT116 cells.

### 3.5. Quercetin Promotes Sp1 Proteasomal Degradation in CRC Cell Lines

Finally, we sought to investigate the molecular mechanism underlying the effects reported here. We first investigated Qc uptake and subcellular distribution taking advantage of its autofluorescence and ability to interact with macromolecules [50,51]. HCT116 cells were exposed to the usual dose of Qc in time-course experiments, PFA-fixed, permeabilized, and analyzed by fluorescence microscopy.

Qc was internalized within a few minutes and progressively accumulated in the nucleus with a homogeneous distribution reaching a peak at 8 h. The signal tended to spread back to the cytosol at 16 h and even more at 24 h of treatment, although it remained mostly nuclear (Figure 6A). The kinetics of the Qc uptake and subcellular distribution are in line with previous reports [50,51,52]; moreover, the prevalent nuclear localization is a general phenomenon, as detected in a number of different cell types, likely underscoring the interaction with macromolecular complexes, including DNA [52,53].

Indeed, it has been reported that many natural compounds, including Qc, can transiently bind to diverse proteins [52]. Specifically, Qc has been shown to interact with Sp1 in pull-down experiments in mesothelioma cells in vitro, preventing the interaction with the binding sites on target gene promoters [54]. The same compounds can trigger a degradation of the interacting proteins through the proteasome, caspases, or other pathways [55,56]. We verified this possibility by exposing HCT116 and HT-29 cells to Qc for 24 h and to MG132, a proteasome inhibitor, for the last 5 h of the treatment. Cells were also treated with the flavonol or MG132 alone as controls. Protein extracts from treated cells were analyzed by a Western blot for Sp1 (Figure 6B). MG132 alone did not affect Sp1, as expected; Qc reduced Sp1, as shown before; the combined treatment (MG132 + Qc) rescued the initial values in both cell lines. The selectivity of the MG132 action was confirmed by the fact that β-Actin and α-Tubulin were not affected, suggesting that Qc instigates the Sp1 degradation through the proteasome.

High-resolution fluorescence microscopy images of treated HCT116 cells confirmed the result. The intensity of the Sp1 fluorescent signal was reduced by about 20% vs. vehicle-treated controls (see Material and Methods). Although obtained by microscopic images, this value is in line with the 30% reduction assessed by a Western blot on extracts from cells treated with Qc for 24 h (Figure 6C).

Finally, we silenced Sp1 by using the RNA interference technology to prove that Qc counteracts its role in tumorigenesis. To this end, we transfected HCT116 and HT-29 cells with three different siRNAs (siRNA1-3) directed against *SP1* mRNA and analyzed cell extracts 48 h later for proteins and RNA. A scrambled siRNA was transfected in both cell lines and used as negative control (Neg_CTR). The Sp1 protein was reduced by all siRNAs in both cell lines; in particular, siRNA1 and siRNA3 lowered Sp1 by about 80–90%, reaching an almost complete disappearance of the protein in the Western blot analysis (Figure 7A). siRNA1 and 3 reduced *SP1* mRNA by about 50–60% in both cell lines as by qRT-PCR (Figure 7B) with a satisfactory variation in silencing efficiency. siRNA2 produced only a modest diminution of Sp1 and was no longer used. The scrambled control did not affect either *SP1* mRNA or protein. We then checked the levels of miR-27a and found they were reduced, along with miR-23a and miR-24-2, confirming that the entire cluster is transcriptionally regulated by Sp1 (Figure 7C). In line with reducing miR-27a levels, its target ZBTB10 increased at both protein and mRNA levels, indicating that Sp1 controls the miR-27a-ZTB10 regulatory axis (Figure 7D,E). *SPRY2* and *PTEN* mRNAs similarly increased, supporting the role of Sp1 in proliferation, phenocopying the results obtained with Qc (Figure 7F,G).

## 4. Discussion

In this study, we investigated the molecular mechanisms underlying the effects of Qc, a flavonol belonging to the large class of polyphenols endowed with beneficial effects on human health both in vitro and in vivo [1,2,3,4,5,6,7,8,9]. We report that Qc acts in a dual mode to counteract the protumorigenic effects of the Sp1-miR-27a axis.

The analysis was carried out in a CRC-derived cell model system represented by HCT116 and HT-29 cells, as colonocytes are one of the first stations where polyphenols introduced with the diet are adsorbed and metabolized. The similar results obtained in two cell lines with distinct genetic and epigenetic landscapes indicate that the effects are general and can be extended to other cell types. The diverse sensitivity to Qc likely reflects these differences and their response to chemotherapeutics [31]. Qc reduces viability in both cell lines at dosages that are not toxic and induces programmed cell death, a process that partially explains the result obtained. The impairment of cell growth is achieved mainly through enhancing Sprouty2, PTEN, and SFRP1, which act as negative regulators of the RAS/ERK, PI3K/AKT, and Wnt/β-catenin cell proliferation pathways, respectively. Sprouty2 reduces ERK1/2 phosphorylation and its nuclear translocation, blocking the growth-stimulating effects mediated by the receptor tyrosine kinase/RAS pathway [40]; PTEN is an AKT phosphatase that reduces p-AKT, impairing cell proliferation [41]; SFRP1 is a soluble form of the frizzled receptor that competes for the Wnt factors, blocking β-catenin phosphorylation, its nuclear translocation, and the activation of cell-cycle progression genes [24]. Thus, Qc reduces the overall cell growth by impairing the major proliferation pathways. Interestingly, the mRNAs of these factors are targets of miR-27a, a microRNA that is overexpressed in HCT116 cells, as well as in CRC in vivo where it acts as an onco-miRNA, greatly contributing to its development. We document here that Qc reduces miR-27a and, consequently, increases the three inversely correlated mRNAs, indicating that miR-27a affects the target mRNAs’ stability and translation, supporting its role as a driver of CRC tumorigenesis.

*MIR27A* is an RNA coding gene and member of a cluster located in an intergenic region on chromosome 19 that includes *MIR23A* and *MIR24-2*. Qc causes a significant reduction not only in all three miRNAs but also of the nascent transcript or pri-miRNA, suggesting a regulation of the entire cluster. Some miRNAs are regulated at the post-transcriptional level, due to the fact that their pri-miRNAs contain a structure, defined as a conserved terminal loop (CTL), that interacts with specific RNA binding proteins (RBP) necessary for their maturation [57,58,59]. In the case of pri-miR-7-1, the RBP HuR and its interacting partner MSI2 bind to the CTL and increase the rigidity of the structure, blocking the cleavage by the microprocessor complex. This facilitates the precursor degradation and prevents a mature miR-7 production [60]. Several natural compounds have been shown to influence this processing [61,62]. Qc dissociates both HUR and MSI2 from the CTL, favoring the processing of the primitive transcript and biogenesis of the mature miR-7 [63]. The pri-miRNA of the miR-23a~27a~24-2 cluster does not bear the CTL and no RBPs or auxiliary factors have been found so far for its processing [57,59], ruling out that Qc acts at the post-transcriptional level; rather, the data suggest a transcriptional regulation.

The miR-23a cluster is expressed at low or very low levels in normal tissues and remarkably increases during the various steps of tumorigenesis [45]. In the normal colonic mucosa, miR-27a is minimally expressed, but already at the adenoma stage, it increases and persists at high levels at all stages of CRC development [22]; in the same cell context, miR-23a has been found to be crucial in the transition from indolent to invasive tumors [64]; moreover, miR-24-2 overexpression contributes to colorectal tumorigenesis, according to some studies [65]. It is therefore conceivable that the overexpression of the entire cluster is under the control of factors associated with tumorigenesis. A comprehensive survey of the promoter identified binding sites for several TFs, in particular for MYC, suggesting a possible regulatory function [66].

However, the promoter region lacks a TATA box, an initiator sequence and a downstream promoter element, while it is enriched in GC elements, putative binding sites for Sp1. This promoter organization is reminiscent of that of housekeeping genes coding for constitutively expressed proteins, genes that, in many cases, are under the transcriptional control of Sp1 (Figure 5). The role that Sp1 plays in the regulation of the miR-23a~27a~24-2 cluster is supported by the observation that it binds to the GC elements located between −600 and −400 bp upstream of the TSS, as their deletion reduces the transcriptional activity of a reporter plasmid [30,43,48].

*SP1* is a member of a family of genes coding for TFs composed by at least nine members but the proteins Sp1–Sp4 are the most studied for their similarity in structure and function [25,26]. Upon binding to GC elements, Sp factors regulate thousands of genes many of which are in common and many others Sp-specific with a cell-context and gene-specific variability likely due to the diverse cellular concentration of each factor. Sps, Sp1 in particular, are overexpressed in tumors and act as transcriptional activators of genes relevant to cell proliferation and tumorigenesis. For these reasons, Sp TFs are termed as nononcogene addiction genes [26].

The interplay of Sp1 with miR-27a delineated so far has an additional level of complexity represented by ZBTB10, a zinc finger protein with a DNA-binding domain but without an activation domain, that recognizes the same GC elements and competes with Sp1 for binding [27]. miR-27a targets *ZBTB10* mRNA and protein, resulting in high Sp1 levels, which stimulate the transcription of the miR-23a~27a~24-2 cluster in a feed-forward positive regulatory loop. Qc reduces both Sp1 protein and mRNA, confirming that this TF regulates its own gene transcription [49] as additionally supported by the ENCODE dataset. Furthermore, the *SP1* promoter is structured similarly to the miR-23a cluster and contains binding sites for other TFs that may interact with and contribute to *SP1* gene transcription [46].

However, Qc, in our CRC cell model, not only interrupts the Sp1-miR-27a-ZBTB10 axis but also triggers Sp1 degradation. Due to its chemical structure, Qc is, in fact, rapidly internalized by the cell and accumulates in the nucleus where it interacts with DNA [50,51,52]. Likely fostered by the interaction with Sp1, Qc instigates its proteasomal degradation of about 20–30%, as supported by the fluorescence microscopy images. SP1 is reported to undergo proteasome-dependent or -independent degradation according to the different tumor cell type [46]. The results reported here that two different CRC cell lines use the same proteasome-dependent pathway suggest that this may be the prevalent one activated by Qc in this tumor type and adds to the interference with the miR-27a-ZBTB10 axis described above.

The central role that Sp1 plays in this scenario is finally supported by the results of its knocking down. The reduction in miR-27a, miR-23a, and miR-24-a further demonstrates that the entire cluster is transcriptionally regulated by Sp1. This lowering, in turn, induces an upregulation of ZBTB10 mRNA and protein that negatively affects both Sp1 protein and *SP1* gene transcription. Then, Sp1’s loss appears not compensated by the other members of the family, especially Sp3 and Sp4, expressed in both cell lines analyzed [34]; if they had, in fact, a fully compensatory role, we would have expected no effects; the result suggests instead that each member plays critical and somehow unique protumoral roles. Alternatively, we have to hypothesize that all Sp factors act in a cooperative manner and the loss of a single member is not prejudicial to a full or partial compensatory rescue by the others [26]. Importantly, the knockdown experiments show that other miR-27a targets are upregulated and impair cell proliferation pathways, phenocopying the effects exerted by Qc. Collectively, Qc acts in a dual mode and exerts its effects by counteracting Sp1 in colorectal cancer.

## 5. Conclusions

In this study, we elucidated the molecular mechanisms underlying the antiproliferative and proapoptotic effects of Qc. Specifically, this flavonol enhances the percentage of apoptotic cells and impairs their proliferation capacity by upregulating Sprouty2, PTEN, and SFRP1, negative modulators of the major growth pathways. Indeed, they are targets of miR-27a, a microRNA that acts as an onco-miRNA and is overexpressed in CRC and other malignancies. Qc not only diminishes the mature miR-27a, but also miR-23a and miR-24-2, the other two members of the miR-23a~27a~24-2 cluster. Interestingly, the nascent transcript or pri-miRNA is also reduced indicating that the entire cluster is regulated, mainly through the transcription factor Sp1 that is over-expressed in tumors. Qc down-regulates Sp1 through the upregulation of ZBTB10, an Sp1 repressor, that is also a target of miR-27a. In addition, Qc forces the Sp1 proteasomal degradation, further reducing the overall Sp1 amount. Sp1’s knocking down reproduces the same results as those of Qc, supporting the central role played by this transcription factor. As a whole, the data provide mechanistic insights into the dual mode of action of Qc and suggest the possibility to use this natural product as an adjuvant in combination with anticancer therapeutic regimens.

## Figures and Tables

**Figure 1 antioxidants-12-01547-f001:**
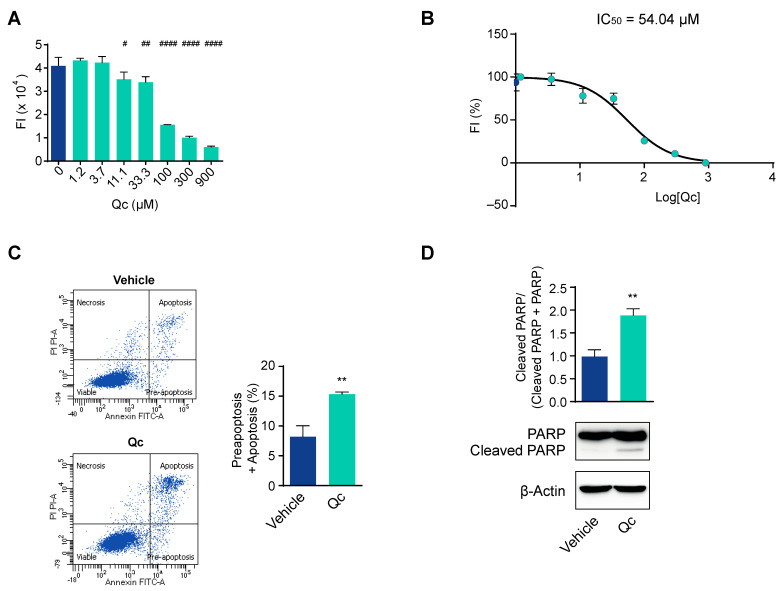
Quercetin influences CRC cells’ viability and apoptosis. (**A**) HCT116 cells were treated with increasing doses of Qc (from 0 to 900 μM) for 24 h and the viability assessed by fluorescence intensity (FI); (**B**) Qc IC_50_ value was calculated in HCT116 cells treated as in (**A**), and FI was reported in percentage; (**C**) Flow cytometry analysis of HCT116 cells treated with 50 µM Qc for 24 h and assayed with the annexin V/propidium iodide test. The lower left quadrants in the figure report viable cells; the upper left, necrotic cells; the upper and lower right indicate apoptotic and preapoptotic cells. The total number of dead cells is shown in the relative histogram and calculated by considering both the preapoptotic and apoptotic cells percentages; (**D**) Western blot analysis of the uncleaved and cleaved form of PARP normalized to the total in HCT116 cells treated with 50 µM Qc or the vehicle alone for 24 h. β-Actin is shown as a protein loading control. Statistical significance is considered when # *p* < 0.05, ## *p* < 0.01, and #### *p* < 0.0001 (ANOVA with Dunnett’s post-test) or ** *p* < 0.01 (*t*-test).

**Figure 2 antioxidants-12-01547-f002:**
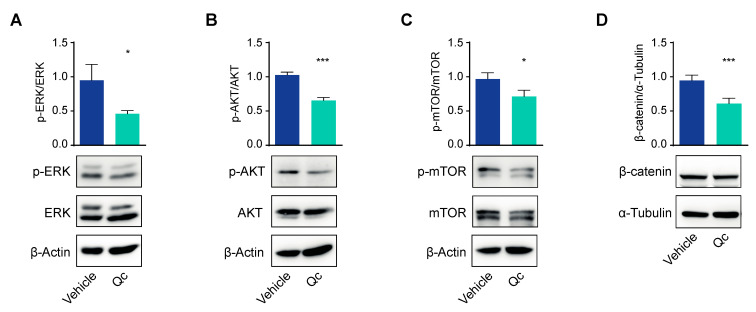
Quercetin impairs the main proliferation pathways in HCT116 cells. Total protein extracts from HCT116 cells treated with 50 µM Qc for 24 h or with the vehicle alone were analyzed by a Western blot for the phosphorylated form of ERK (**A**), AKT (**B**), and mTOR (**C**) or for β-catenin levels (**D**). β-Actin is reported as a protein loading control in (**A**–**C**), α-Tubulin is used for normalization in (**D**). Statistical significance is considered when * *p* < 0.05, and *** *p* < 0.001 (*t*-test).

**Figure 3 antioxidants-12-01547-f003:**
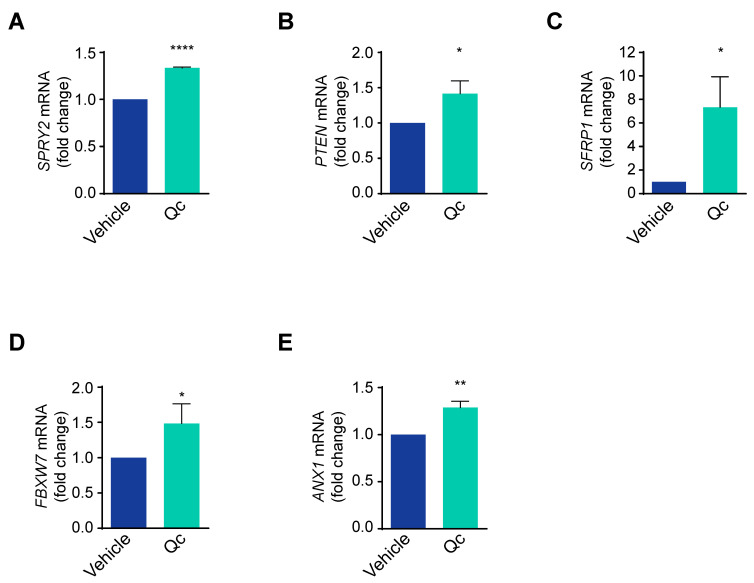
Quercetin upregulates miR-27a targets in HCT116 cells. Total RNA from HCT116 cells treated with 50 µM Qc or with the vehicle alone for 24 h was analyzed by q-RT-PCR for (**A**) *SPRY2*, (**B**) *PTEN*, (**C**) *SFRP1*, (**D**) *FBXW7*, and (**E**) *ANX1* mRNA levels. 18S RNA was used for normalization and is reported as fold change. Statistical significance is considered when * *p* < 0.05, ** *p* < 0.01, and **** *p* < 0.0001 (*t*-test).

**Figure 4 antioxidants-12-01547-f004:**
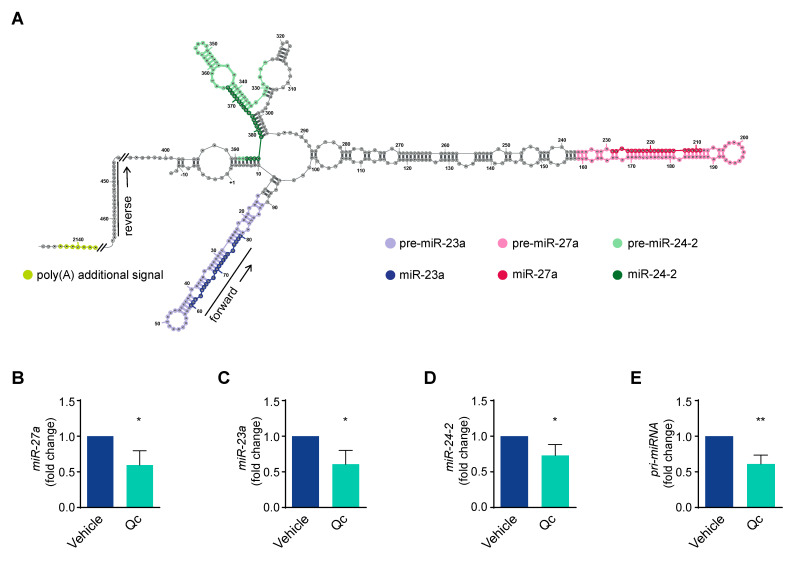
Transcription of the miR-23a~27a~24-2 cluster is modulated by quercetin in HCT116 cells. (**A**) Schematic representation of the miR-27a~23a~24-2 cluster’s nascent transcript (pri-miRNA), obtained by minimizing the free energy within the nucleotide sequence, considering all possible ones. The single miRNAs are highlighted with different colors. The forward and reverse primers used to detect the pri-miRNA fragment are indicated with arrows. Analysis of the mature forms of (**B**) miR-27a, (**C**) miR-23a, (**D**) miR-24-2, and (**E**) pri-miRNA by qRT-PCR in HCT116 cells treated with 50 µM Qc for 24 h or with the vehicle only. U6 RNA was used for normalization and is reported as fold change. Statistical significance is considered when * *p* < 0.05 and ** *p* < 0.01 (*t*-test).

**Figure 5 antioxidants-12-01547-f005:**
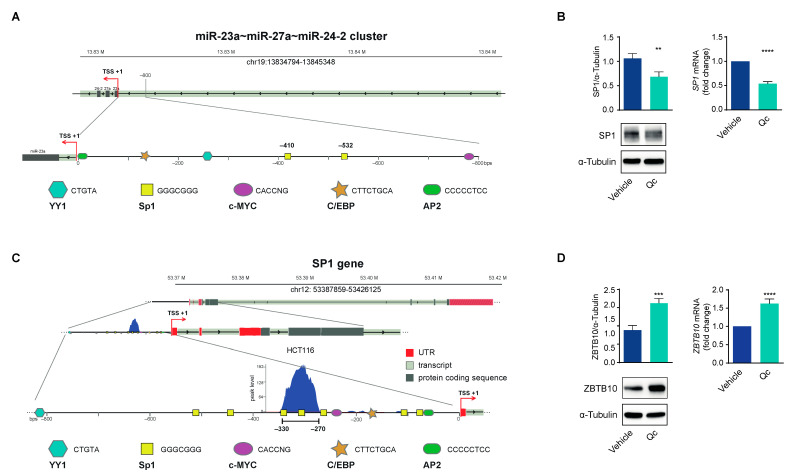
Sp1 and ZBTB10 expression in quercetin-treated HCT116 cells. (**A**) Graphical representation of the putative transcription factors binding sites identified in the miR-23a~27a~24-2 cluster’s 5′ flanking region. (**B**) Sp1 expression in HCT116 cells treated with 50 µM Qc for 24 h or the vehicle alone, analyzed as protein (left panel) or as mRNA (right panel) by a Western blot and qRT-PCR, respectively. α-Tubulin and 18S RNA were used for protein and RNA normalization, respectively. (**C**) Graphical representation of the putative transcription factors’ binding sites identified in the *SP1* gene 5′ promoter region. The blow-up illustrates the selected GC-rich motifs effectively bound by Sp1 reported in the ChIP-Seq dataset (GSM1010902) from the ENCODE Project (ENCSR000BSF) relative to HCT116 cell line. (**D**) ZBTB10 expression in cells treated as in (**B**) analyzed by a Western blot (left panel) and qRT-PCR (right panel), respectively. α-Tubulin and 18S RNA were used for protein and RNA normalization, respectively. Statistical significance is considered when ** *p* < 0.01, *** *p* < 0.001, and **** *p* < 0.0001 (*t*-test).

**Figure 6 antioxidants-12-01547-f006:**
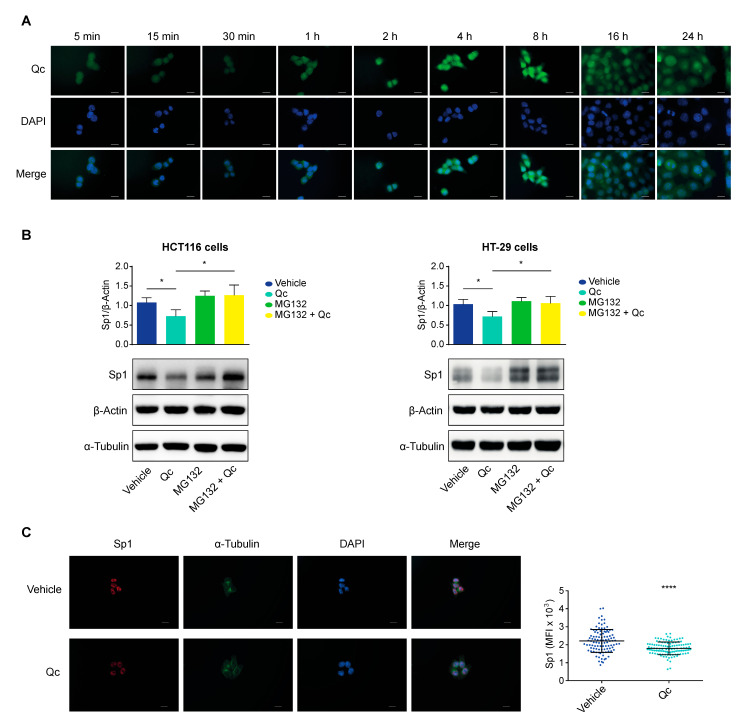
Intracellular localization of quercetin and Sp1 proteasomal degradation in CRC cells. (**A**) Time-course of Qc internalization and subcellular localization in HCT116-treated cells. Qc autofluorescence, shown in green, is detected in the FITC channel, while the nuclei are stained with DAPI and shown in blue (60× objective, ROI (F: 987 × 664, Q: 987 × 664), scale bar 10 μm). (**B**) Western blot analysis of Sp1 in both HCT116 (left panel) and HT-29 (right panel) cells treated with either Qc for 24 h (50 µM and 150 µM, respectively) or with MG132 (10 µM) for the last 5 h of the treatment or in combination. β-Actin was used for protein normalization while α-Tubulin was used as a MG132 negative control. (**C**) Fluorescence microscopy images of Sp1 nuclear localization in HCT116 cells treated or not with Qc for 24 h. Sp1 is shown in red, α-Tubulin in green, and nuclei (DAPI) in blue (60× objective, scale bar 10 μm). The graph on the right illustrates the mean fluorescence intensity (MFI) quantification calculated as reported in the Material and Methods section. All images were taken with the same exposure setting in order to appreciate differences in intensity. Statistical significance is considered when * *p* < 0.05, and **** *p* < 0.0001 (*t*-test).

**Figure 7 antioxidants-12-01547-f007:**
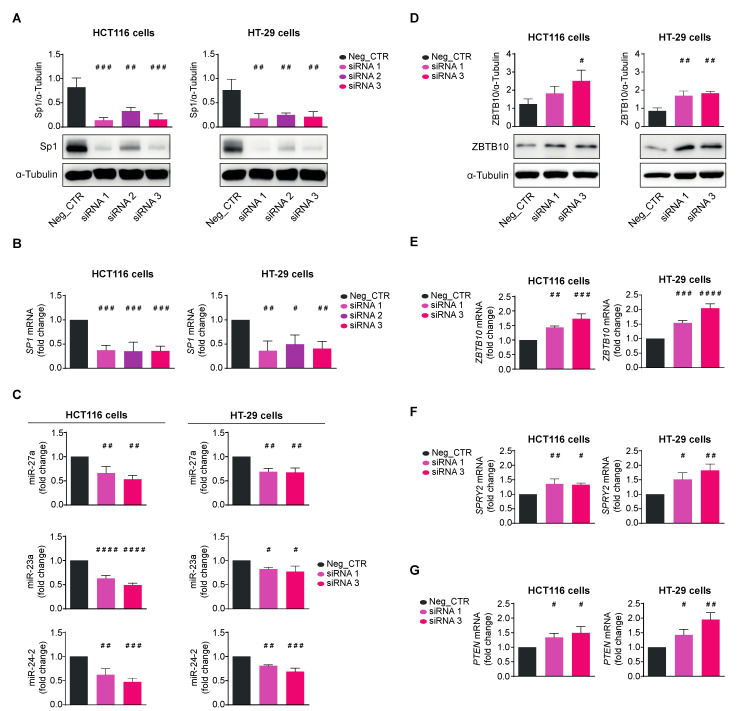
Sp1’s knocking down phenocopies quercetin’s activity in CRC cells. Sp1 expression assessed as protein (**A**) or mRNA (**B**) in extracts from HCT116 (left panels) or HT-29 (right panels) cells transfected for 48 h with a scrambled siRNA (Neg_CTR) or three different siRNAs (siRNA1-3) specific for *SP1* mRNA. α-Tubulin and 18S RNA were used for protein loading and RNA normalization, respectively. (**C**) miR-27a, miR-23a, and miR-24-2 qRT-PCR analysis in HCT116 (left panels) and HT-29 (right panels) cells transfected with the Sp1 siRNA1 or siRNA2 for 48 h compared to Neg_CTR; ZBTB10 protein (**D**) and mRNA (**E**) levels analyzed in HCT116 (left panels) and HT-29 (right panels) cells transfected as in (**C**) and compared to Neg_CTR; (**F**) *SPRY2* and (**G**) *PTEN* RNAs analyzed by qRT-PCR on the total RNA from HCT116 (left panels) and HT-29 (right panels) cells transfected as in (**C**) and compared to Neg_CTR. Statistical significance is considered when # *p* < 0.05, ## *p* < 0.01, ### *p* < 0.001, and #### *p* < 0.0001 (ANOVA with Dunnett’s post-test).

## Data Availability

The data presented in this study are available in the article and from the authors on request.

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
