# Peer review of "Quercetin’s Dual Mode of Action to Counteract the Sp1-miR-27a Axis in Colorectal Cancer Cells"

_antioxidants, 2023, doi:10.3390/antiox12081547_

Round 1

Reviewer 1 Report

This research provides a complex and well-detailed explanation of the anticancer effects of Quercetin (Qc) in colon cancer cells. The author does a commendable job in delineating the mechanism of action of Qc, involving its interaction with miR-27a and its gene cluster, which then upregulates certain negative modulators of proliferation pathways.

The language is technical, as expected in the scientific domain, and the information is dense but logically arranged. This paragraph requires some background knowledge in molecular biology, but for the right audience, it offers valuable insights.

The author has demonstrated a good command of the scientific terminology related to oncology and molecular biology. The flow of the text is logical, detailing how Qc leads to a decrease in the levels of certain miRNAs and affects Sp1, eventually phenocopying the effects of Qc.

One point of critique would be that, while this paragraph is rich in data, it lacks a bit of context for non-expert readers. A brief introductory explanation about Quercetin and its general significance in cancer treatment might help set the stage for the complex mechanism that follows. Furthermore, the text could benefit from shorter sentences, as the current structure may be challenging to follow due to their complexity and length.

Finally, the conclusion adeptly sums up the main point: Qc's potential as a neoadjuvant treatment for cancer. This paves the way for further studies or discussions regarding the therapeutic possibilities of Qc, making this paragraph not only informative but also provocative and forward-looking.

Extensive editing of English language required

Reviewer 2 Report

The concept of the manuscript is interesting in that humans consume fair amount of quercetin in normal diet possibly influencing the appearance of colorectal cancer.

I do have major comments related to the dosage of quercetin used in the study. 

Solubililty of quercetin in aqueous media is fairly limited but the authors used up to 900 micromolar concentrations. Even though such concentration could be reached using organic solvent (DMSO) precipitation will occur in aqueous media. 50 micromolar is approximately the limit. Can the authors provide data, in a form of supplement data, that both concentrations used, i.e. 50 and 150 micromolar, are real?

The other problem is stability of quercetin in aqueous media. The authors applies it for 24 hours in most of their experiments. Again, can they provide data that quercetin in the two concetrations used is not degraded?

Last but not least is the problem with intracellular monitoring of quercetin using autofluorescence. The authors claim that after 5 minutes quercetin accumulates in nucleus. Description of the method used is very brief but the use of paraformaldehyde and Triton must involve wash steps, perhaps more of them. The wash steps may greatly influence the amount and location of quercetin within the cells. The apparent accumulation over time could be combination of diffusion and wash steps. Could a real time monitoring be shown (omitting the fixation and wash) with cell permeable Hoechst used for nuclei labeling? The disappearance of quercetin from nucleus could be related to its stability.

Minor points.

It is not clear what solvent (vehicle) was used to dissolve quercetin.

page 2 line 56 There is a DOI instead of a reference number. Please check.

There are occasional typos in the manuscript, please check for those. E.g. page 5 line 202 "Time-curse", page 14 line 479 "reasobable".

Also the manuscript needs minor english editing.

Reviewer 3 Report

The authors of the manuscript presented the research topic in an interesting way. They planned the experimental work well and described and discussed the results obtained. The work in its present form can be published.
